# Robust Decoupled Motion and Stiffness Control for a Class of Variable-Stiffness Soft Manipulators

*Abstract*— **Variable-stiffness soft manipulators can regulate both motion and compliance, which is essential for robust contact-rich manipulation. However, model-based control remains challenging because motion and stiffness are strongly coupled, the dynamics are highly nonlinear, and tracking performance is sensitive to model uncertainty. We present a nonlinear adaptive cascade controller for electromechanical antagonistic variable-stiffness actuators that achieves closed-loop and decoupled tracking of link motion and joint stiffness. The key idea is to decouple the motion and stiffness channels through a perturbed actuation matrix that remains invertible, and to recover exact stiffness regulation through integral action. Robustness to parametric uncertainty is obtained through online adaptation, while a dead-zone improves tolerance to unmodeled effects such as hysteresis, disturbances, and sensor noise. We validate the method in simulation under 30% parameter error and experimentally on articulated soft robots. Results show accurate motion and stiffness tracking, clear decoupling, and strong robustness, including nearly tenfold lower position error than computed torque control on hardware. This work has been recently published in Transactions on Control Systems Technology.**

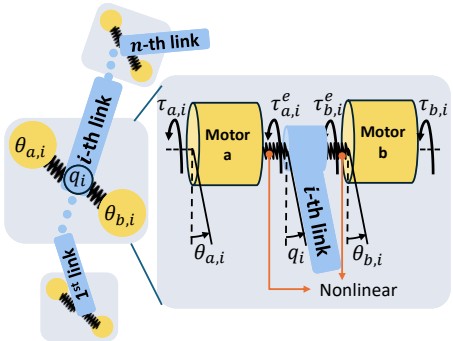

Fig. 1. Illustration of an *ASR* with $n$ rigid links and $n$ compliant joints driven by electromechanical antagonistic VSAs. A nonlinear torque-deflection relation enables real-time stiffness variation. The symbols represent motor positions $\theta_{a,i}$ and $\theta_{b,i}$, link position $q_i$, input motor torques $\tau_{a,i}$ and $\tau_{b,i}$, and elastic torques $\tau_{a,i}^e$ and $\tau_{b,i}^e$ for the $i$-th joint.

## I. INTRODUCTION

Articulated Soft Robots (ASRs), i.e. robots with compliant actuation, that are driven by Variable Stiffness Actuators (VSAs) (Fig 1) have the capability to vary their joints' position and stiffness in real time, thus achieving performance comparable to biological systems [1]. In this domain, antagonistic VSAs are widely employed [2]–[5] as they are characterized by a simpler mechanical design than other VSAs. However, their less complex mechanism comes at the expense of requiring a more sophisticated controller [6].

Foremost, the control task is challenging due to the VSA's highly-nonlinear elastic elements [7] and strong dynamic coupling between motion and stiffness [8], being undesired since varying stiffness can jeopardize precise motion tracking. Various control techniques have been proposed for ASRs with VSAs, ranging from model-free solutions, such as Iterative Learning Control [9], [10], optimal [5], [11] and data-driven approach [12], to model-based ones such as feedback linearization [13], backstepping [14], elastic structure preserving control [15], [16] and observer-based control [17].

We propose a novel strategy for controlling ASRs that achieves simultaneous, independent and closed-loop tracking of link motion (position and velocity) and joint stiffness using a cascade-based control design. We consider here the internal, passive stiffness of a rotational robot joint, physically representing the amount of elastic torque change required to deform the elastic transmission from its current value. Our solution aligns with the philosophy of [8], where both motion and stiffness are controlled in the closed loop through a feedback linearization approach. However, compared to approaches such as feedback linearization [13] and backstepping [14], which require precise knowledge of the model, we use adaptive control strategy [18] to relax the need for the precise parameters knowledge, being motivated by its superior performance when applied on soft robots [19]–[21]. Additionally, we leverage a dead-zone technique to make the system robust also to non-parametric uncertainties. Thus, the first contribution is the design of a control strategy that is robust to uncertainties.

Model-free solutions [9], [10], [12] can successfully deal with the system's high nonlinearity; however, it is challenging to prove the system's stability and they rely on repeated trials or prior identification. Furthermore, strategies such as observer-based control [17] neglect motor dynamics while proving stability. Thus, as a second contribution, we derive a formal stability proof for the overall VSA system, leveraging an adaptive approach to learn in real time and avoid prior identifications.

To address strong motion-stiffness coupling, a recent study approximates the elastic torque with a linear function and achieves simultaneous control of the joint's motion and the VSA's elastic elements pretension [16], while [10] reformulates the model to show that motion and adjusting dynamics can be decoupled under certain assumptions. However, neither solution explicitly controls the joint's stiffness. In contrast, our third contribution enables simultaneous and decoupled control of motion and stiffness dynamics.

**Summary of contributions**. i) We decouple motion and stiffness control by modifying the inverse actuation matrix and incorporating a compensating integral action, laying the groundwork for future research into alternative control

techniques beyond the proposed adaptive strategy. ii) We design a cascade-based controller for a class of VSAs, proving as well stability of the overall closed-loop system. iii) We achieve robustness to uncertainties by leveraging the nonlinear adaptive control theory. By conducting simulation and experimental verification on the one-, two- and three-degree-of-freedom (DoF) VSA-driven ASRs, we prove the efficacy of the proposed control strategy in accurately and independently tracking desired motion and stiffness references.

## II. DYNAMIC MODEL AND PROBLEM STATEMENT

An $n$-DoF robot with compliant joints, driven by electromechanical VSAs, is described by the agonist and antagonist motor position vectors $q_j = (q_{j,1}, \ldots, q_{j,n})^\top$ for $j \in \{a, b\}$, and the link position vector $q = (q_1, \ldots, q_n)^\top$ (Fig. 1). Each joint is actuated by input torques $\tau_{j,i}$ and characterized by transmission deflections $\phi_{j,i} = q_i - q_{j,i}$, generating local elastic torques $\tau_{j,i}^e$ for $i \in \{1, n\}$. The total elastic torque driving the $i$th link is given by their sum. The ASR system dynamics, with a detailed derivation provided in [22], are

$$B(q)\,\ddot{q} + C(q,\dot{q})\,\dot{q} + G(q) + \tau^e(\phi_a, \phi_b) = \tau_{\text{ext}}, \\ B_j\,\ddot{q}_j + D_j\,\dot{q}_j - \tau_j^e(\phi_j) = \tau_j, \quad (1)$$

where $B(q), C(q,\dot{q}) \in \mathbb{R}^{n \times n}$ are the inertia and Coriolis matrices (including viscous friction and centrifugal terms), $G(q) \in \mathbb{R}^n$ is the gravity vector, and $B_j = \text{diag}(b_{i,j})$, $D_j = \text{diag}(d_{i,j})$ are the diagonal motor inertia and damping matrices, $\phi_j = (\phi_{j,1}, \ldots, \phi_{j,n})^\top$ and $\tau_j = (\tau_{j,1}, \ldots, \tau_{j,n})^\top$. The following properties hold:

**Property 1.** $B(q)$ *is symmetric, uniformly positive definite, and bounded for any* $q \in \mathbb{R}^n$, *which implies* $B^{-1}(q)$ *always exists and is bounded* [18].

**Property 2.** *Matrix* $\dot{B}(q) - 2C(q,\dot{q})$ *is skew symmetric, which implies* $v^\top(\dot{B}(q) - 2C(q,\dot{q}))v = 0$, *for all* $v$.

**Property 3.** *The link regressor matrix,* $Y(q,\dot{q},\ddot{q})$ *(cf. Sec. 7.3.2 in [23]) and the block-diagonal motor regressor matrices,* $Y_j = \text{diag}_i\{(\dot{q}_{j,i}\ \ddot{q}_{j,i})\} \in \mathbb{R}^{n \times 2n}$, *for* $j \in \{a, b\}$, *allow expressing the following:*

$$B(q)\,\ddot{q} + C(q,\dot{q})\dot{q} + G(q) = Y(q,\dot{q},\ddot{q})\,\pi, \\ B_j\,\ddot{q}_j + D_j\,\dot{q}_j = Y_j(\dot{q}_j,\ddot{q}_j)\,\pi_j,$$

*where* $\pi_j = (d_{j,1}, b_{j,1}, \cdots, d_{j,n}, b_{j,n})^\top$ *and* $\pi$ *is a suitable link parameter vector.*

**Property 4.** *The system* (1) *has elastically decoupled joints. Thus, its elastic potential energy can be expressed for each joint individually as* $U_i^e = \sum_j U_{j,i}^e(\phi_{j,i})$ *and the total elastic torque* $\tau_i^e$ *depends only on the local deflections, i.e.* $\tau_i^e = \tau_{a,i}^e(\phi_{a,i}) + \tau_{b,i}^e(\phi_{b,i})$ [10]. *Moreover, each term of the joint stiffness vector* $\sigma = (\sigma_1, \cdots, \sigma_n)^\top$ *is defined as* $\sigma_i = \partial\tau_{a,i}^e/\partial\phi_{a,i} + \partial\tau_{b,i}^e/\partial\phi_{b,i}$ [22], [24] *and its time derivative is*

$$\dot{\sigma}_i = (\partial^2\tau_{a,i}^e/\partial\phi_{a,i}^2)\,\dot{\phi}_{a,i} + (\partial^2\tau_{b,i}^e/\partial\phi_{b,i}^2)\,\dot{\phi}_{b,i}. \quad (2)$$

A further property helps in achieving decoupled control, by ensuring that both motion and stiffness dynamics share the same control inputs, i.e. local elastic torques:

**Property 5.** *Elastic torques within a VSA are such that*

$$\partial^2\tau_{j,i}^e/\partial\phi_{j,i}^2 = \nu_{j,i}\,\tau_{j,i}^e, \ \ for \ j \in \{a, b\}, \quad (3)$$

*where* $\nu_{j,i}$ *are suitable scalars.*

Meeting this requirement, i.e. ensuring that each $\tau_{j,i}^e$ solves (3), guarantees that $\tau_i^e$ is an exponential function of the local deflections. By the linearity of (2), this result translates also to the $i$th joint stiffness' dynamics. VSAs that satisfy the foregoing property are indeed quite common [4], as they are inspired by the behavior of biological muscles. Numerous examples include commercial VSAs with sine-hyperbolic elastic torque [25] and VSAs with the exponential springs [26]–[30].

Having said that, the full dynamics of the robot, including that of the joint stiffness, is:

$$B(q)\,\ddot{q} + C(q,\dot{q})\,\dot{q} + G(q) + \tau_a^e(\phi_a) + \tau_b^e(\phi_b) = 0, \quad (4a)$$
$$\dot{\sigma}_i = \nu_{a,i}\,\tau_{a,i}^e(\phi_{a,i})\,\dot{\phi}_{a,i} + \nu_{b,i}\,\tau_{b,i}^e(\phi_{b,i})\,\dot{\phi}_{b,i}, \ \forall i, \quad (4b)$$
$$B_j\,\ddot{q}_j + D_j\,\dot{q}_j - \tau_j^e(\phi_j) = \tau_j, \quad (4c)$$

The above system is underactuated as $q_j$, for $j \in \{a, b\}$ are collocated variables that are directly controlled via the inputs $\tau_j$, while $q$ and $\sigma$ are non-collocated variables and only indirectly controlled via the elastic torques $\tau_j^e$.

To this respect, the following final property is assumed:

**Property 6.** *The mapping* $\tau_{i,j}^e = f(q_i - q_{i,j})$ *between elastic torques and motor positions is smooth and invertible, which implies that one can always write* $q_{j,i} = q_i - f^{-1}(\tau_{j,i}^e)$.

One can now readily introduce the following problem:

**Problem 1.** *Given a VSA-driven robot with nominal model* (4), *design a* robust *control law for the input torque* $\tau_a$ *and* $\tau_b$ *that allow desired motion and stiffness signals,* $q_d$, $\dot{q}_d$ *and* $\sigma_d$, *be asymptotically tracked in a decoupled way. Robustness is intended with respect to parametric uncertainty in the vectors* $\pi$ *and* $\pi_j$, *and decoupling refers to the ability to track* $q_d$ *and* $\dot{q}_d$ *irrespective of* $\sigma_d$ *and vice-versa.*

## III. DECOUPLED NONLINEAR ADAPTIVE CONTROL OF ELECTROMECHANICALLY-DRIVEN ASR

Achieving decoupled control of link motion and joint stiffness is challenging due to system underactuation and strong coupling, while ensuring accuracy is further complicated by parametric uncertainties affecting both link and motor dynamics. This section presents the proposed strategy, which reaches robustness through a nonlinear adaptive control approach [18] and addresses underactuation via a cascade control scheme (Fig. 2). Specifically, an inner loop regulates motor positions to generate elastic torques (cf. Property 6), while an outer loop uses these torques to track user-specified position, velocity and stiffness references. Property 3 is used below and parameter vectors are assumed constant or vary slowly.

**Theorem 1** (Inner-loop Nonlinear Adaptive Control)**.** *Given the actuator model in* (4c) *and desired signals* $q_{j,d} : [0, \infty) \to \mathbb{R}^n$, *with* $q_{j,d} \in C^2$, *for* $j \in \{a, b\}$, *global and robust*

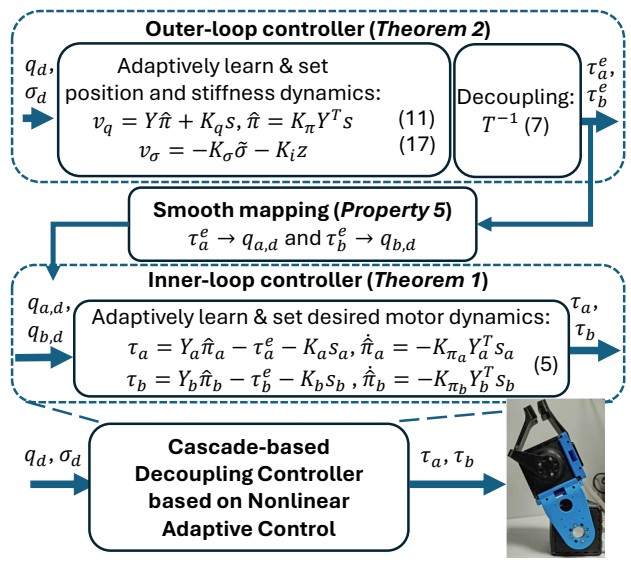

Fig. 2. Block scheme of the proposed nonlinear adaptive control based control algorithm for the decoupled motion and stiffness control of ASR.

*asymptotic convergence of the tracking errors $\tilde{q}_j = q_j - q_{j,d}$ and $\dot{\tilde{q}}_j = \dot{q}_j - \dot{q}_{j,d}$ is guaranteed by the adaptive control law*

$$\tau_j = Y_j(q_j, \dot{q}_j, \ddot{q}_{j,d})\,\hat{\pi}_j - \tau_j^e(\phi_j) - K_j s_j \,,$$
$$\dot{\hat{\pi}}_j = -K_{\pi_j} Y_j(q_j, \dot{q}_j, \ddot{q}_{j,d})^\top s_j \,, \tag{5}$$

*where $s_j = \dot{q}_j - \dot{q}_{j,r}$ and $\dot{q}_{j,r} = \dot{q}_{j,d} - \Lambda_j(q_j - q_{j,d})$, with $\Lambda, K_j, K_{\pi_j} \succ 0$ are control gains, and $Y_j = (\ddot{q}_{j,1,r}, \dot{q}_{j,1}, \cdots)$.*

*Proof:* The proof is provided in Appendix.

**Theorem 2** (Outer-loop Adaptive Control of Link Motion and Joint Stiffness). *Given an ASR with dynamics as in (4a) and (4b) and desired vector signals $q_d, \sigma_d : [0, \infty) \to \mathbb{R}^n$, with $q_d \in \mathcal{C}^1$, global and robust asymptotic convergence of the tracking errors, $\tilde{q} = q - q_d$, $\dot{\tilde{q}} = \dot{q} - \dot{q}_d$ and $\tilde{\sigma} = \sigma - \sigma_d$, is guaranteed by the adaptive control law for the elastic torque*

$$\begin{pmatrix} \tau_a^e \\ \tau_b^e \end{pmatrix} = T^{-1}(q, \dot{q}) \begin{pmatrix} Y(q, \dot{q}, \dot{q}_r, \ddot{q}_r)\,\hat{\pi} + K_q\,s \\ -K_\sigma\,\tilde{\sigma} - K_i\,z \end{pmatrix} , \tag{6a}$$

$$\dot{\hat{\pi}} = K_\pi\,Y(q, \dot{q}, \dot{q}_r, \ddot{q}_r)^\top s \,, \tag{6b}$$

*where $s = \dot{q} - \dot{q}_r$, $\dot{q}_r = \dot{q}_d - \Lambda(q - q_d)$, with $\Lambda \succ 0$, $\dot{z} = \tilde{\sigma}$, $Y$ is the link regressor, $\hat{\pi}$ is a link parameter estimate, $K_q$, $K_\sigma$, $K_i$, and $K_\pi$ are control gains, and*

$$T = \begin{pmatrix} \mathbb{I}_n & \mathbb{I}_n \\ diag_i\{\nu_{a,i}\dot{\phi}_{a,i} + \Delta_i\} & diag_i\{\nu_{b,i}\dot{\phi}_{b,i} - \Delta_i\} \end{pmatrix} , \tag{7}$$

*where $\Delta_i$ is a small constant ensuring $T$ is invertible.*

*Furthermore, given the parameter estimation error $\tilde{\pi} = \pi - \hat{\pi}$, a Lyapunov control function and its Lie derivative, ensuring uniform global asymptotic stability, are*

$$V = \tfrac{1}{2}\left(s^\top B(q) s + \tilde{\sigma}^\top \tilde{\sigma} + z^\top K_i z + \tilde{\pi}^\top K_\pi^{-1}\tilde{\pi}\right) , \tag{8}$$
$$\dot{V} = -s^\top K_q\,s - \tilde{\sigma}^\top K_\sigma\,\tilde{\sigma} \,.$$

*Proof:* The proof is provided in Appendix.

**Remark 1.** *The proposed methodology accommodates diverse control techniques, as evident from (15). The decoupling*

*matrix $T$ enables independent design of motion and stiffness controllers via $v_q$ and $v_\sigma$, facilitating future exploration of alternative approaches such as model predictive control, optimal control [31], recurrent neural networks [32], adaptive neural networks [33], or a baseline computed torque control [34] where $v_q = -C(q, \dot{q})\dot{q} - G(q) + B(q)(\Lambda\tilde{\dot{q}} + K_q s - \ddot{q}_d)$.*

## IV. METHOD APPLICATION AND EVALUATION

### A. Derivation of the Controller

*1) Link dynamics:* Defining the robot's configuration vector as $q = (q_1, q_2)^\top$ and joint stiffness vector as $\sigma = (\sigma_1, \sigma_2)^\top$, the robot dynamics follow (4a) and (4b). The link dynamics is in [19] (cf. page 8). Moreover, from (14), the actuation matrix is $T = \{T_{ik}\}_{ik}$, with $T_{11} = T_{12} = \mathbb{I}_2$, $T_{21} = \text{diag}_i\left\{a_a^2\dot{\phi}_{a,i} + \Delta_i\right\}$, and $T_{22} = \text{diag}_i\left\{a_b^2\dot{\phi}_{b,i} - \Delta_i\right\}$, where $\dot{\phi}_{j,i} = \dot{q}_i - \dot{q}_{j,i}$.

*2) Actuation:* For motor dynamics (4c), the inertia and damping matrices are $B_a = \text{diag}_i\{b_{a,i}\}$, $B_b = \text{diag}_i\{b_{b,i}\}$, $D_a = \text{diag}_i\{d_{a,i}\}$, and $D_b = \text{diag}_i\{d_{b,i}\}$. According to [25], the $i$th VSA device applies an elastic torque and sets a joint stiffness as

$$\tau_i^e = k_a \sinh(a_a\phi_{a,i}) + k_b \sinh(a_b\phi_{b,i}),$$
$$\sigma_i = a_a k_a \cosh(a_a\phi_{a,i}) + a_b k_b \cosh(a_b\phi_{b,i}), \tag{9}$$

where $k_a$, $k_b$, $a_a$, and $a_b$ are experimentally identified spring constants. Expanding sinh as $\tau_i^e = k_a/2\left(e^{a_a\phi_{a,i}} - e^{-a_a\phi_{a,i}}\right) + k_b/2\left(e^{a_b\phi_{b,i}} - e^{-a_b\phi_{b,i}}\right)$ shows Property 5 is met. Using (9), motor positions are $q_a = q - \text{asinh}\left(\tau_a^e/k_a\right)/a_a$ and $q_b = q - \text{asinh}\left(\tau_b^e/k_b\right)/a_b$, ensuring Property 6.

*3) Regressors and parameters:* The link-side regressor and the parameter vector are in [19] (cf. page 8).

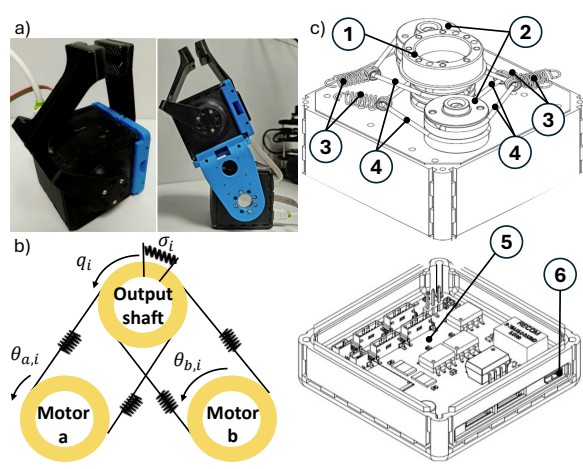

Fig. 3. Experimental setup: (a) one- and two-DoF hardware setups for validation; (b) a bidirectional antagonistic setup where $\theta_{a,i}$ and $\theta_{b,i}$ are motor positions, $\sigma_i$ is stiffness, and $q_i$ is the $i$-th joint's output position. Joint motion is achieved by rotating both motors in the same direction, while stiffness is adjusted by moving them oppositely; (c) an exploded VSA view showing (1) output shaft, (2) pulleys, (3) four nonlinear springs, (4) four tendons, (5) a PCB with a Cypress PSoC 3 microcontroller handling motor control and position sensing via AS5045 encoders (12-bit), and (6) a micro-USB port for communication and firmware deployment.

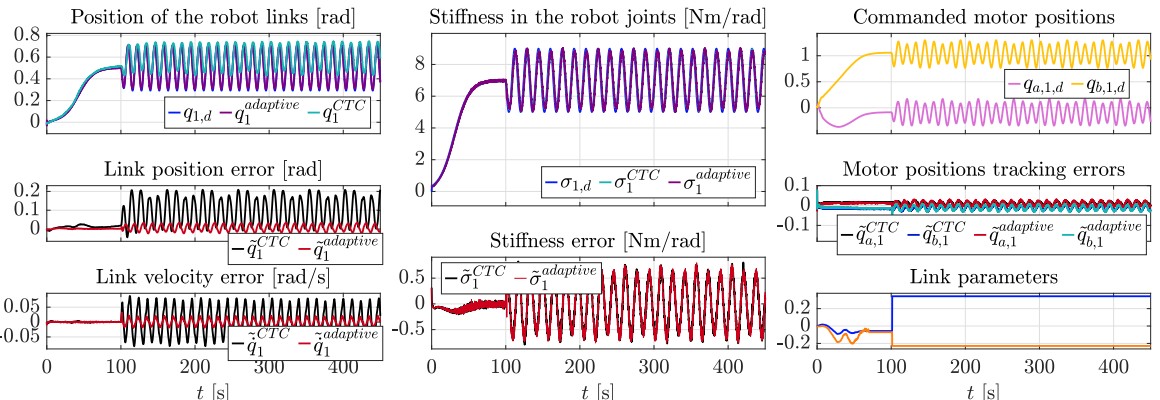

Fig. 4. Experimental run of a 1-DoF ASR using computed torque control and the proposed adaptive approach for motion tracking. The computed torque control (CTC) leads to significant position and velocity tracking error due to the lack of robustness to uncertainties. With identical stiffness control in both cases, stiffness reliably reaches the desired setpoint and tracks the trajectory with bounded error. Commanded motor positions evolve smoothly in both cases.

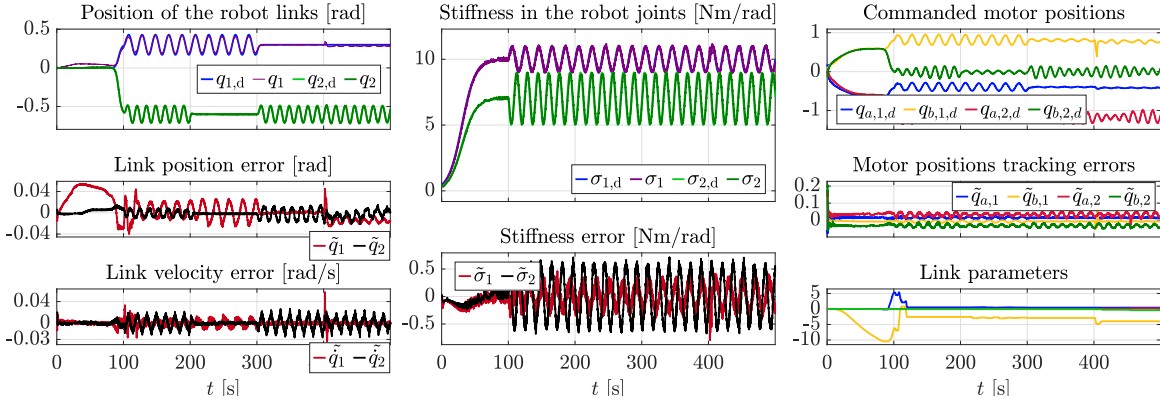

Fig. 5. Experimental run of a two-DoF ASR using the proposed control. Joint positions, velocities and stiffnesses are accurately tracked, with bounded parameters and smooth commands. The leftmost figure shows decoupling, as link positions remain stable despite stiffness and other link variations. At 400s, adding a load to the second link caused a brief tracking error, which was quickly compensated within the dead-zone bounds.

## B. Experimental Validation

The proposed control approaches are finally validated using an ASR with VSAs in an antagonistic setup as in Fig. 3a.

*1) Hardware and software setup:* The robot is an articulated arm with two revolute joints, each driven by an antagonistic VSA *qbmove Maker Pro* [25]. This bidirectional antagonistic actuator allows each motor to apply both positive and negative torques (Fig. 3b). The main mechanical and electronic components are shown in Fig. 3c. We show a 1-DoF and 2-DoF qbmove setup results; whole experimental validation is shown in the video (link).

*2) Experimental results for the 1-DoF ASR:* The 1-DoF robot's dynamic model is $(I_1 + m_1 l_1^2) \ddot{q}_1 + m_1 g l_1 \sin q_1 + \tau_a^e + \tau_b^e = 0$, with the regressor matrix, $Y = (\ddot{q}_{1,r}, \sin q_1)$, and the unknown parameters $\pi = (I_1 + m_1 l_1^2, m_1 g l_1)^\top$. The proposed control is compared to the baseline computed torque control (CTC). Fig. 4 shows that adaptive control reduces position tracking error nearly tenfold by adapting parameters. Quantitatively, it achieves a root mean square error of 0.016 rad for position and $0.01 \frac{\text{rad}}{\text{s}}$ for velocity tracking, significantly outperforming the CTC (0.1 rad and $0.045 \frac{\text{rad}}{\text{s}}$).

*3) Experimental results for the 2-DoF ASR:* We extend the analysis to a 2-DoF ASR using two *qbmove Maker Pro* actuators (Fig. 3a) to demonstrate decoupling and tracking.

Joint positions remain constant in certain segments, while stiffness and the other joint's position vary. The first joint's stiffness is set higher to support the structure. Fig. 5 confirms precise tracking and successful decoupling, as the position and velocity remain unaffected by stiffness changes or the other joint's movement. Also, we apply a disturbance at 400 s by placing an irregular object that weighs 0.25 kg on the end-effector. Parameters converge to the constant, but adapt when disturbed, keeping the error within $|s| \leq \eta = 0.1$.

## V. DISCUSSION AND CONCLUSION

This abstract presents a nonlinear adaptive control strategy for robust and decoupled tracking of motion and stiffness in electromechanical ASRs. Our solution applies to the button-pushing task from [35], requiring accurate motion tracking to reach a button and precise stiffness regulation: too little stiffness prevents the button from being pushed, while too much may cause damage to the robot or the environment. It also holds promise for surgical robotic tasks such as needle insertion. Moreover, its independent motion and stiffness control enables intuitive replication of human arm movement. Future work will focus on extending to different VSAs, exploring alternative control laws and control variables, and addressing uncertainties in actuator mapping.

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

## APPENDIX

**Proof of Theorem 1.** The proof shows the convergence of the motor tracking errors despite parametric uncertainty of the vectors $\pi_j$ involved in Prop. 3. It assumes a dynamic control law is used that is based on motor parameter estimates $\hat{\pi}_j$, which are continuously updated. To this purpose, for each agonist and antagonist motor, consider the estimation errors of motor parameters $\tilde{\pi}_j = \hat{\pi}_j - \pi_j$, for $j \in \{a, b\}$ and the Lyapunov candidate $V_j = \frac{1}{2}(s_j^\top B_j s_j + \tilde{\pi}_j^\top K_{\pi_j}^{-1} \tilde{\pi}_j)$. In the Lie derivative $\dot{V}_j = s_j^\top B_j \dot{s}_j + \dot{\hat{\pi}}_j^\top K_{\pi_j}^{-1} \tilde{\pi}_j = s_j^\top B_j (\ddot{q}_j - \ddot{q}_{j,r}) + \dot{\hat{\pi}}_j^\top K_{\pi_j}^{-1} \tilde{\pi}_j$, replace $B_j \ddot{q}_j$ with its expression derived from (4c) and use Property 3, which leads to

$$\dot{V}_j = s_j^\top \left(\tau_j + \tau_j^e(\phi_j) - D_j \dot{q}_j - B_j \ddot{q}_{j,r}\right) + \dot{\hat{\pi}}_j^\top K_{\pi_j}^{-1} \tilde{\pi}_j$$
$$= s_j^\top \left(\tau_j + \tau_j^e(\phi_j) - Y_j(\dot{q}_j, \ddot{q}_{j,r})\,\pi_j\right) + \dot{\hat{\pi}}_j^\top K_{\pi_j}^{-1} \tilde{\pi}_j,$$

with $\pi_j = (b_{j,1}, d_{j,1}, \cdots, b_{j,n}, d_{j,n})^\top$. Plugging the first laws in (5) gives $\dot{V}_j = s_j^\top (Y_j \tilde{\pi}_j - K_j s_j) + \dot{\hat{\pi}}_j^\top K_{\pi_j}^{-1} \tilde{\pi}_j \leq -s_j^\top K_j s_j + (s_j^\top Y_j(\dot{q}_j, \ddot{q}_{j,r}) + \dot{\hat{\pi}}_j^\top K_{\pi_j}^{-1})\tilde{\pi}_j$, and selecting the parameter adaptation laws as in the second group of (5) results in the negative semidefinite function $\dot{V}_j = -s_j^\top K_j s_j$. To show that $s_j \to 0$ as $t \to \infty$, it suffices to prove that $\dot{V}_j \to 0$. Since $V_j \succ 0$, Barbalat's lemma ensures $\dot{V}_j \to 0$ if $\dot{V}_j$ is uniformly continuous. This holds if $\ddot{V}_j = -2s_j^\top K_j \dot{s}_j$ is bounded, requiring both $s_j$ and $\dot{s}_j$ to be bounded. The boundedness of $s_j$ follows from $V_j \succ 0$ and its non-increasing nature ($\dot{V}_j \prec 0$), ensuring $V_j$ remains bounded. To establish $\dot{s}_j$ bounded, note that $q_{j,d}$ are bounded. The closed-loop dynamics $\ddot{\tilde{q}}_j + K_{d,j} \dot{\tilde{q}}_j + K_{p,j} \tilde{q}_j = 0$ imply that $\ddot{\tilde{q}}_j$ is bounded, given that both $\tilde{q}$ and $\dot{\tilde{q}}_j$ are bounded. Thus, $\dot{s}_j$ is also bounded.

With inner loop convergence achieved, Property 6 ensures that the desired local elastic torques are reached, allowing us to proceed with the derivation of the outer-loop controller.

**Proof of Theorem 2.** *(1. Compound dynamics derivation).* The link acceleration $\ddot{q}$ and stiffness dynamics can be factorized in terms of convenient inputs. Specifically, using (4) and having defined $v_q = (\mathbb{I}_n, \mathbb{I}_n)(\tau_a^{e\top}, \tau_b^{e\top})^\top$, one obtains

$$\ddot{q} = -B(q)^{-1}(C(q,\dot{q})\dot{q} + G(q) + \tau_a^e + \tau_b^e) = \\ = -B(q)^{-1}(C(q,\dot{q})\dot{q} + G(q) + v_q). \tag{10}$$

Defining then $v_\sigma = (\Gamma_a, \Gamma_b)(\tau_a^{e\top}, \tau_b^{e\top})^\top$, with $\Gamma_j = \text{diag}_i\left\{\nu_{j,i}\,\dot{\phi}_{j,i}\right\}$, one obtains the stiffness dynamics

$$\dot{\sigma} = \begin{pmatrix} \nu_{a,1}\dot{\phi}_{a,1}\,\tau_{a,1}^e + \nu_{b,1}\dot{\phi}_{b,1}\,\tau_{b,1}^e \\ \vdots \\ \nu_{a,n}\dot{\phi}_{a,n}\,\tau_{a,n}^e + \nu_{b,n}\dot{\phi}_{b,n}\,\tau_{b,n}^e \end{pmatrix} = v_\sigma. \tag{11}$$

*(2. Derivation of control inputs $v_q$ and $v_\sigma$).* Expand the Lie derivative of the Lyapunov candidate $V = \frac{1}{2}(s^\top B(q)s + \tilde{\sigma}^\top\tilde{\sigma} + \tilde{\pi}^\top K_\pi^{-1}\tilde{\pi})$:

$$\dot{V} = s^\top B(q)\dot{s} + \frac{1}{2}s^\top \dot{B}(q)s + \tilde{\sigma}^\top\dot{\tilde{\sigma}} + \tilde{\pi}^\top K_\pi^{-1}\dot{\tilde{\pi}} = \\ = -s^\top(B(q)\ddot{q}_r + C(q,\dot{q})\dot{q} + G(q) + v_q) + \\ + \frac{1}{2}s^\top \dot{B}(q)s + \tilde{\sigma}^\top v_\sigma + \tilde{\pi}^\top K_\pi^{-1}\dot{\tilde{\pi}}.$$

Plugging the factorized link acceleration from (10) into $\dot{s} = \ddot{q} - \ddot{q}_r$, and that of the stiffness dynamics from (11), then using the skew-symmetry of $\frac{1}{2}\dot{B}(q) - C(q,\dot{q})\dot{q}$ (Property 2) leads to $\dot{V} = -s^\top(\alpha(q) + v_q) + \tilde{\sigma}^\top v_\sigma + \tilde{\pi}^\top K_\pi^{-1}\dot{\tilde{\pi}}$, where $\alpha(q) = B(q)\ddot{q}_r + C(q,\dot{q})\dot{q}_r + G(q)$ and the assumption of a constant or slowly changing $\pi$ has been done. Using Property 3 and the relation $\pi = \hat{\pi} - \tilde{\pi}$, one can rewrite $\alpha(q) = Y\hat{\pi} - Y\tilde{\pi}$, for suitable $Y = Y(q,\dot{q},\dot{q}_r,\ddot{q}_r)$. Then, the Lie derivative of $V$ is written as $\dot{V} = -s^\top(Y\hat{\pi} + v_q) + \tilde{\sigma}^\top v_\sigma + \tilde{\pi}^\top(K_\pi^{-1}\dot{\hat{\pi}} + Y^\top s)$, where the equivalence $s^\top Y\tilde{\pi} = (s^\top Y\tilde{\pi})^\top = \tilde{\pi}^\top Y^\top s$ has been used. Now, one can now force $\dot{V}$ be equal the desired function $\dot{V}_d = -s^\top K_q s - \tilde{\sigma}K_\sigma\tilde{\sigma}$. Solving the equation $\dot{V} = \dot{V}_d$ for $v_q$, $v_\sigma$, and $\dot{\hat{\pi}}$ gives

$$v_q = Y(q,\dot{q},\dot{q}_r,\ddot{q}_r)\hat{\pi} + K_q\,s, \quad v_\sigma = -K_\sigma\,\tilde{\sigma}, \tag{12}$$

and the parameter adaptation law (6b). As $s = \dot{\tilde{q}} + \Lambda\tilde{q} \to 0$, so it happens with $\tilde{q}$ and $\dot{\tilde{q}}$. Also, $\hat{\pi}$ remain bounded.

Moreover, plugging the control input $v_\sigma$ from (12) into the stiffness dynamics (11), we obtain $\dot{\sigma} = -K_\sigma\tilde{\sigma}$, by which each $\sigma_i \to \sigma_{d,i}$, for all $i$, with convergence speed of $k_{\sigma,i}$.

*(3. Mapping the control inputs $v_q$ and $v_\sigma$ to elastic torques).* The control laws $v_q$ and $v_\sigma$ are mapped to the corresponding elastic torque signals by solving the system

$$T'\begin{pmatrix} \tau_a^e \\ \tau_b^e \end{pmatrix} = \begin{pmatrix} v_q \\ v_\sigma \end{pmatrix}, \quad \text{with } T' = \begin{pmatrix} \mathbb{I}_n & \mathbb{I}_n \\ \Gamma_a & \Gamma_b \end{pmatrix} \tag{13}$$

whose solvability requires $T'$ be invertible. As all blocks in $T'$ are square and with the same sizes and the in-diagonal blocks, $\mathbb{I}_n$ and $\Gamma_b$, commute with each other, one gets via Sylvester's rule [36]:

$$|T'| = |\mathbb{I}_n\Gamma_b - \mathbb{I}_n\Gamma_a| = \text{diag}_i\left\{\nu_{b,i}\dot{\phi}_{b,i} - \nu_{a,i}\dot{\phi}_{a,i}\right\} = \\ = \Pi_{i=1}^n(\nu_{b,i}\dot{\phi}_{b,i} - \nu_{a,i}\dot{\phi}_{a,i}),$$

showing $T'$ is not invertible if $\nu_{a,i}\dot{\phi}_{a,i} = \nu_{b,i}\dot{\phi}_{b,i}$ for any $i$.

*(4. Perturbation ensuring invertibility of $T'$).* To overcome the invertibility issue, an equation system as in (13) is solved,

replacing $T'$ with a perturbed yet invertible $T$. In determining how to obtain $T$, note that, while the primary goal is still to control both link motion and joint stiffness, motion tracking is typically more critical than stiffness tracking in most applications. Therefore, only the rows of $T'$ related to the stiffness subsystem are perturbed. The choice

$$T = T' + \begin{pmatrix} 0_n & 0_n \\ \Delta & -\Delta \end{pmatrix} = \begin{pmatrix} \mathbb{I}_n & \mathbb{I}_n \\ \Gamma_a + \Delta & \Gamma_b - \Delta \end{pmatrix}, \tag{14}$$

where $\Delta = \text{diag}_i\{\Delta_i\}$ and all $\Delta_i$ are (small) constants, serves the purpose. Observing that the upper-left block in $T$, $\mathbb{I}_n$, is invertible and assuming that its Schur complement in $T$ (cf. [37], page 44), $M = \Gamma_b - \Delta - (\Gamma_a + \Delta)\mathbb{I}_n^{-1}\mathbb{I}_n = \Gamma_b - \Gamma_a - 2\Delta$, is also invertible, one gets

$$T^{-1} = \begin{pmatrix} \mathbb{I}_n + M^{-1}(\Gamma_a + \Delta) & -M^{-1} \\ -M^{-1}(\Gamma_a + \Delta) & M^{-1} \end{pmatrix}.$$

The (unique) solution of the perturbed system is

$$(\tau_a^{e\top}, \tau_b^{e\top})^\top = T^{-1}(v_q^\top, v_\sigma^\top)^\top. \tag{15}$$

Plugging (15) into (13) and subtracting the unperturbed solution yields the calculation error $\epsilon = (\epsilon_q^\top, \epsilon_\sigma^\top)^\top$ due to perturbation:

$$\epsilon = T'T^{-1}\begin{pmatrix} v_q \\ v_\sigma \end{pmatrix} - \begin{pmatrix} v_q \\ v_\sigma \end{pmatrix} = \left(\begin{pmatrix} \mathbb{I}_n & 0_n \\ T_{21} & T_{22} \end{pmatrix} - \mathbb{I}_{2n}\right)\begin{pmatrix} v_q \\ v_\sigma \end{pmatrix},$$

with $T_{21} = \Gamma_a + (\Gamma_a - \Gamma_b)M^{-1}(\Gamma_a + \Delta)$ and $T_{22} = \Gamma_b M^{-1} - \Gamma_a M^{-1}$. To express $\epsilon$ in terms of $\Delta$, we factorize $T_{21}$, by adding and subtracting $2\Delta M^{-1}(\Gamma_a + \Delta)$, leading to $T_{21} = \Gamma_a - 2\Delta M^{-1}(\Gamma_a + \Delta) - MM^{-1}(\Gamma_a + \Delta) = -\Delta(\mathbb{I}_n + 2M^{-1}(\Gamma_a + \Delta))$. Similarly, adjusting $T_{22} - \mathbb{I}_n$ by adding and subtracting $2\Delta M^{-1}$ gives $T_{22} - \mathbb{I}_n = (\Gamma_b - \Gamma_a - 2\Delta)M^{-1} + 2\Delta M^{-1} - \mathbb{I}_n = MM^{-1} + 2\Delta M^{-1} - \mathbb{I}_n = 2\Delta M^{-1}$. Thus, the perturbation-induced calculation error is

$$\epsilon = \Delta\begin{pmatrix} 0_n & 0_n \\ -(\mathbb{I}_n + 2M^{-1}(\Gamma_a + \Delta)) & 2M^{-1} \end{pmatrix}\begin{pmatrix} v_q \\ v_\sigma \end{pmatrix},$$

which expands to

$$\epsilon_q = 0, \quad \epsilon_\sigma = \Delta(2M^{-1}v_\sigma - (\mathbb{I}_n + 2M^{-1}(\Gamma_a + \Delta))v_q).$$

Since $v_q$ stabilizes motion tracking, it is accurately computed, ensuring $e \to 0$. However, $v_\sigma$ retains a steady-state calculation error, $\epsilon_\sigma \to Av_\sigma$ and $A = 2\Delta M^{-1}$. As $\Delta$ and $M$ are diagonal, $A = \text{diag}_i\left\{2\Delta_i/(\nu_{b,i}\dot{\phi}_{b,i} - \nu_{a,i}\dot{\phi}_{a,i} - 2\Delta_i)\right\}$. When close to the singularity of $T'$, i.e., $\nu_{a,i}\dot{\phi}_{a,i} \approx \nu_{b,i}\dot{\phi}_{b,i}$, the $i$th diagonal entry of $A$ tends to $-1$, making $A \to -\mathbb{I}_n$. Away from singularity, $A$ remains bounded but nonzero, leading to a small yet persistent steady-state stiffness error.

*(5. Compensation by integral action and final version of the controller).* Following the idea in [38], an integral term is added to a new Lyapunov candidate $W$ so as to reestablish the stiffness convergence. Let $z$ be a newly introduced variable with dynamics $\dot{z} = \tilde{\sigma}$ and consider the Lyapunov candidate

$$W = \frac{1}{2}\tilde{\sigma}^\top\tilde{\sigma} + \frac{1}{2}z^\top K_i z, \tag{16}$$

where $K_i$ is a control gain. The Lie derivative of $W$,

$$\dot{W} = (\tilde{\sigma}^\top, z^\top)(v_\sigma^\top, (K_i\tilde{\sigma})^\top)^\top, \tag{17}$$

is made negative semidefinite and equal to

$$\dot{W} = \left(\tilde{\sigma}^\top, z^\top\right) \begin{pmatrix} -K_\sigma \tilde{\sigma} - K_i z \\ K_i \tilde{\sigma} \end{pmatrix} = -\tilde{\sigma}^\top K_\sigma \tilde{\sigma},$$

by choosing the control law $v_\sigma = -K_\sigma \tilde{\sigma} - K_i z$, with $K_\sigma$ a control gain. The stiffness tracking error convergence is ensured despite $\dot{W}$ being only negative semi-definite as per LaSalle's invariance principle [39]. Namely, imposing the condition $\tilde{\sigma} = 0$ be invariant for the controlled system also implies $\dot{\sigma} = 0$; plugging the two conditions into $\dot{\sigma} = -K_\sigma \tilde{\sigma} - K_i z$, finally leads to $z = 0$. Finally, direct computation of the overall Lyapunov control function and its Lie derivative yields (8).

**Stability Analysis.** Analyzing the stability of the adaptive cascade control scheme requires writing model (4) in cascade form. To this aim, define $x_1 = \left(q^\top, \dot{q}^\top, \sigma^\top\right)^\top$, $x_2 = (q_a^\top, \dot{q}_a^\top, q_b^\top, \dot{q}_b^\top)^\top$, $\theta_1 = \pi$, $\theta_2 = (\pi_a^\top, \pi_b^\top)^\top$, which yields $\dot{x}_1 = f_1(t, x_1, \theta_1) + g(t, x, \theta_1, \theta_2)$, $\dot{x}_2 = f_2(t, x_2, \theta_2)$, where $t \in \mathbb{R}_{\geq 0}$ and $f_1$, $f_2$ and $g$ are Lipschitz in the state and piecewise continuous in time for $\theta_1$ and $\theta_2$. According to [40], the above model is Uniformly Semi-globally Practically Asymptotically Stable (USPAS) for all parameter sets $\Theta_1 \subset \mathbb{R}^{m1}$ and $\Theta_2 \subset \mathbb{R}^{m2}$, if: (a) the dynamics of $x_2$ is USPAS on $\Theta_2$; (b) $\dot{x}_1 = f_1(t, x_1, \theta_1)$ is USPAS and the solutions of $x_1$ converge irrespectively of the origin's attractive neighborhood; (c) the interconnection term $g$ is as $|g(t, x, \theta_1, \theta_2)| \leq G(|x|)$ where $G : \mathbb{R}_{\leq 0} \to \mathbb{R}_{\leq 0}$ is a non-decreasing function; (d) the model's solutions are uniformly bounded.

As for (a), Th. 1 proves the uniform exponential stability of the closed-loop motor dynamics, which is a stricter condition than being USPAS. As for (b) and (d) can be validated by following the procedure in [40] and via the Lyapunov function in Th. 2. As for (c), Property 1 allows proving that the entries of $T$ are bounded, namely all $\Gamma_j$ are bounded by the lemma in [40], which stands due to the triangle inequality $|\Gamma_j| = \nu_j |\dot{q} - \dot{q}_j| \leq \nu_j |\dot{q}| + \nu_j |\dot{q}_j|$.