# OpenReview forum: "Robust Decoupled Motion and Stiffness Control for a Class of Variable-Stiffness Soft Manipulators"
_IEEE.org/ICRA/2026/Workshop/Manipulation_Robustness — ICRA 2026_

### Official Review · Reviewer_XsF1 · 2026-05-02
**A well‑engineered but actuator‑specific approach for robust decoupled motion/stiffness control in antagonistic soft manipulators**

**Rating:** 8
**Confidence:** 4

**Review:**

Strengths:

1. The paper proposes a cascade adaptive controller that effectively decouples link motion and joint stiffness in variable-stiffness soft manipulators, with a stability proof and clear experimental validation.

2. Robustness to parametric uncertainty and unmodeled effects is achieved through online adaptation and a dead‑zone, demonstrating nearly tenfold lower position error than a computed‑torque baseline on hardware.

Weaknesses:

1. The key Property 5 (exponential elastic torque) is restrictive; the perturbed actuation matrix trick introduces potential steady‑state stiffness error that is compensated by integral action, adding complexity.

2. The experimental evaluation is limited to 1‑DoF and 2‑DoF setups, and the impact of sensor noise, hysteresis, and more diverse disturbances is only partially explored.

3. As the core contribution has already appeared in a journal, the workshop paper does not offer substantially new content, reducing the novelty for a workshop audience.

Overall comments:

This work presents a sound, well‑engineered control solution for antagonistic variable‑stiffness actuators. Its strength lies in the decoupling of motion and stiffness through a perturbed actuation matrix and in the robust adaptive law that learns parameters online. While the heavy reliance on an exponential spring characteristic makes the method less general than one might hope for “soft robots” at large, it does cover many devices inspired by biological muscle, and the stability analysis is rigorous within that scope.

---

### Decision · Program_Chairs · 2026-05-21

Accept